# A Gold Nanoparticle Bioconjugate Delivery System for Active Targeted Photodynamic Therapy of Cancer and Cancer Stem Cells

**DOI:** 10.3390/cancers14194558

**Published:** 2022-09-20

**Authors:** Onyisi Christiana Didamson, Rahul Chandran, Heidi Abrahamse

**Affiliations:** Laser Research Centre, Faculty of Health Sciences, University of Johannesburg, Johannesburg 2028, South Africa

**Keywords:** gold nanoparticle, bioconjugate, drug-delivery system, photodynamic therapy, cancer and cancer stem cells

## Abstract

**Simple Summary:**

Cancer stem cells (CSCs) play a crucial role in tumor development, metastasis, therapy resistance, and relapse due to their self-renewal and proliferative potential. In this review, we summarized the application of gold nanoparticle (AuNPs) bioconjugates in enhancing the efficiency of photodynamic therapy (PDT) in cancer and CSCs. We also highlighted the challenges facing the translation of AuNPs application in clinical settings.

**Abstract:**

Cancer stem cells (CSCs), also called tumor-initiating cells, are a subpopulation of cancer cells believed to be the leading cause of cancer initiation, growth, metastasis, and recurrence. Presently there are no effective treatments targeted at eliminating CSCs. Hence, an urgent need to develop measures to target CSCs to eliminate potential recurrence and metastasis associated with CSCs. Cancer stem cells have inherent and unique features that differ from other cancer cells, which they leverage to resist conventional therapies. Targeting such features with photodynamic therapy (PDT) could be a promising treatment for drug-resistant cancer stem cells. Photodynamic therapy is a light-mediated non-invasive treatment modality. However, PDT alone is unable to eliminate cancer stem cells effectively, hence the need for a targeted approach. Gold nanoparticle bioconjugates with PDT could be a potential approach for targeted photodynamic therapy of cancer and CSCs. This approach has the potential for enhanced drug delivery, selective and specific attachment to target tumor cells/CSCs, as well as the ability to efficiently generate ROS. This review examines the impact of a smart gold nanoparticle bioconjugate coupled with a photosensitizer (PS) in promoting targeted PDT of cancer and CSC.

## 1. Introduction

Cancer is a deadly malignancy that continues to be the leading cause of sickness and death globally. Cancer accounted for about 9.9 million mortality and 19.2 million new cases worldwide in 2020. It is projected that by 2040 the global cancer burden will increase to about 29–37 million new cases of cancer. Some of the top leading types of cancer with increased cancer-associated death are lung, colorectal, stomach, liver, breast, esophageal, pancreas, and prostate [1]. Cancer emerges because of defects in the genes regulating cellular homeostasis between cell proliferation and cell death. These defects in the regulatory genes result in an imbalance in the cell cycle and apoptosis, leading to excessive cell growth, the collapse of cellular tissue function, tumor invasion of neighboring cells, and consequently metastasis and disease progression. Cancer cells have the potential not to undergo apoptosis and are characterized by aggressive cell proliferation compared to non-cancerous cells [2].

In recent years, much success has been achieved in cancer treatment. The primary treatment modalities for cancer treatments are surgery, chemotherapy, and radiotherapy [3]. The treatment of choice depends on the tumor type, location, histological type, size, stage, and the presence and level of metastasis. In invasive carcinoma, neoadjuvant chemoradiation therapy (nCRT) and then surgery is the treatment options [4,5,6]. However, the response rate is very poor, with over 50% of tumor with partial response and 20% with no response completely. No doubt, current conventional treatments for cancer have reduced the burden of cancer to some degree; however, treatment failure and disease progression are still huge concerns and main challenges to be tackled.

Emerging reports propose that the failure of conventional therapy such as surgery, chemotherapy, and radiotherapy to eliminate cancer is collectively due to treatment resistance and progression of metastasis. Cancer metastasis and treatment resistance work hand in hand to invade their surrounding tissues and, upon establishing evasion, continue to metastasize and migrate to other tissues [7]. As a result, a subpopulation of cancer called cancer stem cells CSCs can escape therapeutic pressure and establish metastatic invasion by coordinating several factors such as signaling pathways, DNA/damage repair programs, anti-apoptotic potential, and metabolic modification. These factors have been shown to promote stemness, metastasis, drug resistance, recurrence, and poor prognosis [7,8]. Therefore, identifying and targeting these common factors that facilitate metastasis and therapeutic resistance should offer a promising approach to eradicating tumor and enhancing anticancer treatment.

## 2. Cancer Stem Cells

Cancer stem cells (CSCs) are a small subset of cells implicated in cancer development, growth, spread, treatment resistance, and relapse. Cancer stem cells arise from the dysregulation of normal tissue stem cells during differentiation. Under chemotherapy pressures, differentiated cancer cells encounter cell death, while CSCs are resistant to treatment pressure, preventing apoptosis. These residual CSCs can re-establish their presence, leading to clinical cancer relapse [9,10]. Genetic alterations and epigenetic modification in normal stem cells often lead to cancer and are promoted by their self-renewal capability [8,11]. Cancer stem cells are recognized and characterized by the following features, uncontrollable proliferation, self-renewal, massive differentiation, immune evasion, and treatment resistance. Cancer stem cells have been implicated in various cancers following treatment failure [12,13,14,15,16].

Cancer stem cells are characterized by the incredible ability to develop into any cells present in the body, high resistance to treatments and the potential to form new tumor in appropriate animal hosts. The CSCs consist of two phenotypes, the proliferative and quiescent/dormant states, as demonstrated by their capability to undergo differentiation and de-differentiation. The CSCs with proliferative phenotypes differentiate from normal cancer cells, which form the bulk of the tumor tissue mass and are more responsive to treatment. While the quiescent phenotype CSCs have de-differentiation potentials and are thought to be the driver of drug resistance, metastasis, and relapse [17,18]. Therefore, it is crucial to have in-depth knowledge and understanding of the origin and attributes of CSCs to tackle the challenge of drug resistance, relapse, and metastasis.

### 2.1. Origin and Theory of Cancer Stem Cells

In 1994 Lapidot and coworkers first identified a subpopulation of CD34+CD38−- CSCs in acute myeloid leukemia (AML). The expression of CD34 and CD38 were employed as markers to identify premature cells in the normal bone marrow. The transplant of CD34+CD38− subpopulation into severe combined immuno-deficient (SCID) mice developed more tumor growth than CD34+CD38+ or CD34− subpopulations [19]. In 1997, Bonnet and Dick further confirmed the evidence of these subpopulations CSCs. They discovered that these subpopulations of cells have distinct characteristics from the primary cancer cells but have the same properties as normal stem cells from which the term CSCs was derived [20]. Subsequent reports have identified the presence of CSCs, with their unique features and cellular recognition, in various forms of solid tumors, including esophagus [21,22], colon [23], melanoma [24], breast [25], prostate [26] and brain cancer [27].

The precise mechanism of the development of CSC development is quite not clear. However, theoretical models have explained their presence in cancer tissue: The classical or stochastic model proposes that any somatic cells have the inherent potential to develop into CSCs caused by genetic or epigenetic alteration, as represented in Figure 1A [14,28]. The cancer stem cell or hierarchical model states that CSCs arise from adult stems or adult progenitor cells through mutation or cell/tumor cell differentiation that acquires stem-like features in de-differentiation. The initiating tumor cell regenerates during differentiation forming both CSCs and normal tumor cells. The normal tumor cells form the cells in the bulk tumor through cell division, as shown in Figure 1B [28,29]. The plasticity/stemness phenotype model states that cancer cells are comprised of both CSCs and non-CSC phenotypes and can interconvert into each other due to their plasticity characteristics and EMT. In addition, any single fittest and survival cancer cell can initiate a new tumor Figure 1C [14,28,30].

### 2.2. Biomarkers of Cancer Stem Cell and Isolation

Prompt diagnosis is very crucial for efficient treatment and monitoring of cancer. During cancer development, the cellular components undergo different alterations. These alterations can help in tumor detection, tracking tumor growth and prognosis. Cancer stem cells have unique molecules within or outside the cell known as markers and are employed for their characterization and isolation. Various markers have been recognized in identifying CSCs, as depicted in Table 1. These markers are used to determine treatment outcomes. The expression of some of these markers is associated with a poor prognosis. In addition, overexpression of the CSCs markers is implicated in the various stages of tumor initiation, differentiation level, the extent of invasion, and distant metastasis [9,30].

Several techniques for identifying and isolating CSCs from primary tumor and cancer cell lines have been developed for functional characterization and effective therapeutic targeting. Tumor formation by CSCs in nonobese diabetic/severe combined immune-deficient (NOD/SCID) mice has been the standard technique for identifying this subpopulation of cells. Notwithstanding, various in vitro methods, such as fluorescence-activated cell sorting (FACS) or magnetic cell separation, are presently exploited to identify CSCs based on the expression of specific extracellular or intracellular recognition molecules [9]. A vast insight into these markers and their cellular role could help design novel treatments. Therefore, targeting CSCs is very important for cancer therapy. The limitations of present cancer treatment modalities to attack CSCs resulted in the urgent quest for new treatment approaches for cancer and cancer stem cells with reduced side effects.

### 2.3. Cancer Stem Cell Niche

The CSC niche is a unique tumor microenvironment (TME) that promotes the self-renewal and continuous proliferation of CSCs. The niche consists of extrinsic signals (extracellular matrix (ECM) proteins, signaling factors, networks of cytokines, growth factors, physicochemical factors, etc.) and stromal cells (fibroblastic cells, immune cells, endothelial and perivascular cells, etc.) located close to CSCs. The distinct physicochemical factors and the various elements in the niche synergically work together to support tumorigenesis, metastasis, and resistance to anticancer treatment [31,32]. With regards to the complex features of CSCs, conventional anticancer therapy can eliminate less aggressive cancer cells resulting in the reduction in the tumor size. However, they cannot eliminate CSCs, leading to relapse and survival of CSCs. The residual CSCs, through the support of the CSCS niche, develop and re-establish new tumor [33,34].

### 2.4. Mechanism of Treatment Resistance in Cancer Stem Cells

It is not entirely understood the mechanism CSCs employ to escape treatment pressure. Meanwhile, some fundamental mechanisms have been established, such as (i) overactivation of detoxification enzymes aldehyde dehydrogenase (ALDH), (ii) improved DNA repair efficiency, (iii) overstimulation of drug resistance proteins, (iv) upregulation of anti-apoptotic proteins (Bcl-2, Bcl-xL, Mcl-l, Bcl-w), (v) hyperactivation of drug efflux pumps (P glycoprotein 1, adenosine triphosphate-binding cassette superfamily G member 2 (ABCG2)) and (vi) alterations in vital signaling molecules have been observed to confer resistance to anticancer agents in CSCs [8,18]. Hence, targeting these mechanisms offers a potent strategy for eliminating CSCs. Furthermore, multiple signaling cascades such as Hedgehog, Wnt/beta-catenin, JAK-STAT3, Notch and Hippo pathways are upregulated in cancer and are found to initiate and maintain CSCs. These pathways promote self-renewal, uncontrol propagation, and differentiation and cause treatment failure in cancer expression [9,30]. In addition, abnormal regulation of micro-RNAs, autophagy, hypoxia and stimulation of epithelial-mesenchymal transition (EMT) can individually or in synergy trigger the quiescent CSCs through abnormal activation of signaling pathways to the emergence of cancer relapses and therapeutic failure in cancer [10]. Emerging reports have shown that targeting these signaling pathways with specific inhibitors could reduce tumor cells’ stemness [9,10,31].

The mechanisms highlighted above pose a severe threat to CSCs treatment. Presently, conventional cancer treatments cannot eliminate metastasized tumor and are ineffective at targeting CSCs. Though there has been advancement in searching for a novel strategy for CSCs elimination, implementing an effective technique to eliminate resistant tumor cells is still a concern. A significant limitation to overcome is the development of target-specific treatment approaches aimed at CSCs alone without affecting the normal stem cells, as some cell receptors and signaling pathways are expressed in CSCs and normal stem cells [8]. In addition, the degree of the adverse effects and low survival rate associated with the current treatment necessitates the call for an alternative treatment modality.

## 3. Photodynamic Therapy

Photodynamic therapy (PDT) is a non-surgical treatment modality employed for various disorders not effectively eliminated by conventional therapeutic measures, such as cancer. Photodynamic therapy is seen as an emerging treatment option, demonstrating high efficiency with few side effects compared with the several unwanted effects associated with conventional treatment. Photodynamic therapy consists of three vital components, a photosensitizer (PS), light and molecular oxygen. The PS (a non-toxic agent) is administered and localized in the tumor cells, which is then triggered by light of a specific wavelength that corresponds to its absorption parameters. In the presence of molecular oxygen, the activated PS, through cascades of photochemical reactions, initiates the selective destruction of tumor and adjacent blood vessels [42,43].

The photochemical reactions are classified into two, type I and type II. In type I reaction, reactive oxygen species (such as superoxide radical anion (O_2_^•−^), hydrogen peroxide (H_2_O_2_), and hydroxyl radicals (HO^•^)) are formed when the excited-state PS either gain or loses an electron, often from the cellular substrate with less molecular oxygen concentrations. The ROS generated causes stress to the treated tissues and, finally, cell death. While type II reaction involves the transfer of energy from a triplet excited state in the presence of molecular oxygen to generate singlet oxygen (^1^O_2_). The singlet oxygen reacts with a myriad of cellular substrates, stimulating severe oxidative stress and eventually cell damage. Type II reaction is known as the classical pathway of PDT.

Though type I and type II reactions rely on oxygen, the subcellular oxygen level in the tumor microenvironment regulates the reaction type [42,43]. The initiation of cell death is dependent on the subcellular localization of PS and the production of sufficient ROS and cytotoxic singlet molecular oxygen. A mitochondria-localized PS would possibly initiate an apoptotic pathway, while PS internalized in the lysosomes stimulate cell death via cleavage of BID and a necrotic reaction upon a high PDT dose. [44]. Furthermore, the efficiency of PDT mainly relies on the type of PSs. An effective PS should be chemically pure, have a light absorption peak within the therapeutic window (600 nm–800 nm), have a selective preference for tumor cells, possess the potential to generate a long-lasting triplet excited state, remain inactive in the absence of light, easily cleared from the body, and have amphiphilic property. Some common PSs employed in PDT are hematoporphyrin, methylene blue, chlorins, bacteriochlorin, curcumin, phthalocyanines, and hypericin [45].

### 3.1. Photodynamic Therapy and Route of Administration

The route of PDT administration is determined by the PS, site of treatment and the light delivery device. Effective PSs should have appropriate administration routes (such as systemic, oral, or topical). Efficient PDT depends on the ease of administering the treatment light distinctly to the entire target tissue [46]. The most common routes of administration employed in preclinical and clinical applications of PDT are intravenous, topical, and oral administration of PSs. Photosensitizer administered topically and orally is generally employed due to easy administration, safety, and less expensive, while intravenous and intravesical administration exhibit reduced skin phototoxicity in vivo. However, more administration routes with excellent outcomes are emerging, such as intraperitoneal, intra-arterial, and intratumoral injections. They are more beneficial when compared to intravenous injections due to their ability to deliver the PS right at the site of the tumor, shorten the drug-light interval, minimize the adverse effects and increase survival time [47].

### 3.2. Photosensitizer OptimumInjection Dosage for PDT

The optimum injection dosage of any PS for PDT is regulated by multiple factors such as the type of tumor, location of the tumor, the PS pharmacokinetics, tissue oxygenation, light fluence, and patient variability (intra and inter) [48], as well as the formulation type [49]. For instance, porfimer sodium (Photofrin) is formulated in phosphate-buffered saline (PBS) and given through intravenous injection at a dose of 2 mg/kg). Thereafter within two days after injection, irradiation is applied with a light dose of 100–200 J/cm at 630 nm [50]. Porfimer sodium is recommended for clinical application against early lung cancer, esophageal cancer, and high-grade Barrett’s esophagus [51].

Temoporfin (Foscan) also known as m-tetrahydroxyphenylchlorin (mTHPC) is formulated in ethanol and propylene glycol in the ratio of 2:3 volume by volume. Temoporfin is injected intravenously at a dose of 0.15 mg/kg, and the drug-light interval of one to four days. It is irradiated with a light dose of 20 J/cm at 652 nm. The potency of Temoporfin is remarkably higher than that of porfimer sodium at similar PS dose and light dose. The advantage of Temoporfin is that it allows the use of a lower drug dose and a shorter irradiation time. Temoporfin has been approved for clinical application of head cancer and neck cancer [49,50]. Talaporfin sodium is formulated in a water-soluble medium and administered via intravenous route at a drug dose of 1 mg/kg with a drug-light interval of 0.25–4 h. Talaporfin sodium is rapidly cleared and has less skin photosensitivity [50].

### 3.3. Evaluation of Photodynamic Therapeutic Efficiency

An integral part of PDT to targeted tissue is dose/treatment evaluation, and this is vital for efficient response. Presently, the generation of singlet oxygen (^1^O_2_) during PDT is generally considered to be the main indicator of PDT dose efficiency administered. Therefore, much interest is directed toward effective detection and measurement of ^1^O_2_ generation and its relation to responsive treatment outcome [52]. Various dose measurement (dosimetry) techniques can be applied for PDT dose evaluation, such as implicit dosimetry, explicit dosimetry, direct dosimetry, and biophysical/biological tissue response monitoring [48,52].

Implicit dosimetry measures the loss of fluorescence (photobleaching) caused by ^1^O_2_ during treatment. Various studies have shown a remarkably relationship between PS photobleaching and treatment efficacy. Explicit dosimetry evaluates the PS concentration, light fluence, and tissue oxygenation. Explicit dosimetry offers a simplified approach to PDT dosimetry considering the three core elements of PDT. However, it is very challenging to measure all three elements accurately in a clinical setting. Direct dosimetry measures the exact ^1^O_2_ produced during therapy, which is considered to be the amount inducing cellular damage. This is achieved by measuring its near-infrared (NIR) luminescence at 1270 nm. Direct singlet oxygen luminescence dosimetry (SOLD) is preferred as it eliminates the challenges associated with implicit and explicit dosimetry [48,52].

Biophysical and biological tissue response investigation can be utilized as a dosimetric strategy. This involves measuring vascular obstruction, treatment-mediated tissue damage, and vascular flow evaluation by laser diffuse correlation spectroscopy or Doppler. New emerging techniques for PDT dosimetry are on the rise. The most common dosimetry techniques applied in the clinic are the imaging modalities, such as ultrasound, dynamic contrast-enhanced (DCE), computed tomography (CT) and DCE magnetic resonance imaging (MRI) [53]. In vivo findings have shown that ultrasound-guided photoacoustic imaging and non-invasive tumor vascular flow evaluation in the course of treatment can be employed to predict treatment response [54].

### 3.4. Benefits and Limitations of PDT

The main benefit of employing PDT is that it promotes preferential cell damage and has limited effects on neighboring non-cancerous tissues. Photodynamic therapy has shown to be an efficient anti-tumor therapy and improves prognosis. Its effectiveness is observed when in combination with other treatment modalities and can be applied as a first resort for preneoplastic lesions, early staged tumor, and as an independent treatment for palliative therapy. In addition, it is less invasive, retains the anatomic and functional structure of many cells, reduces side effects, absence of drug resistance, and permits continual treatment owing to its minimal toxicity [42,55].

Although PDT has some excellent advantages over conventional treatment, it is still faced with some drawbacks that limit its full application in the standard management plan for cancer. A major setback observed is the limited light/PS penetration in deep-seated and metastasized tumor, thereby rendering PDT ineffective. Furthermore, tumor hypoxia, a disparate feature of solid tumor, greatly renders PDT ineffective as it depends on molecular oxygen to produce ROS. Finally, the hydrophobic nature of the PSs results in systemic aggregation, uneven delivery, decreased cellular localization, and lack of specificity [42,55,56,57]. Several tumor-targeting approaches have evolved to overcome these setbacks and enhance PDT efficiency in clinical settings. Recent evidence has demonstrated the advantages of nanotechnology-mediated PDT in overcoming the limitations associated with PDT efficiency [58,59,60].

## 4. Nanotechnology for Enhancing Cancer Therapy

The emergence of nanotechnology has ushered a new trajectory for improved cancer therapy. Nanotechnology is a multidisciplinary field emerging from the crosslink of biology, physics, chemistry, optics, digital analysis, and material science. It involves the design, synthesis, characterization, modification, and application of different groups of tiny particles known as nanoparticles (NPs). The evolution of nanotechnology in medicine has made significant advancements in the diagnosis, drug design, drug-delivery system and treatment of different diseases, including cancer, due to their unique characteristics [61,62]. Their unique properties are mainly due to their high loading capacity, improved electrical conductivity, superparamagnetic activity, spectral shift of optical absorption, and superior fluorescence features. In addition, they exist in different shapes and sizes allowing for easy functionalization and providing potential strategies for the design of cancer and CSC treatment. Their features are also associated with delivering drugs to targeted tissue, escaping biological barriers, and interacting with cellular and intracellular molecules [61].

There are different types of NPs, and includes organic (e.g., micelles, liposomes, dendrimers, ferritin, and polymers NPs), inorganic (e.g., Au, Fe, Al, Ag, Zn, and metal oxides), and carbon-based NPs (e.g., graphene and Fullerenes) [60]. Among these, nanoparticle gold (Au) NPs have drawn much attention due to their excellent physicochemical and electro-optical features. Based on these features, they are employed in various applications such as drug-delivery systems to targeted cells, biosensors, cancer diagnosis and therapy [62,63].

## 5. Gold Nanoparticles and Drug Delivery

Gold NPs have been well investigated as promising NPs for several applications in biomedicine, such as diagnosis, drug delivery, cancer therapy, and treatment of various diseases. This is due to distinct features such as superior compatibility and stability in biological systems, flexible surface functionalization/modification, excellent optical properties for efficient imaging, electroconductivity features, chemical inertness, and different sizes and shapes [62,64]. The surface chemistry of AuNPs has been well leveraged for the attachment of targeting molecules, imaging tags, and therapeutic agents with several multiple applications, especial as a delivery system. AuNPs could be transported to cancer tissues as drug nanocarriers via passive or active targeting approaches.

In passive targeting, NPs penetrate tumor tissue through the porous blood vessels via the enhanced permeation and retention (EPR) effect. Based on this concept, highly porous tumor blood vessel promotes increased movement of AuNPs into the cancer tissue. In addition, damaged lymphatic drainage prevents the clearance of the AuNPs in tumor, leading to enhanced retention. These attributes are mainly due to the increased growth rate of cancer and the disintegration of the blood and lymph vessels [65]. Active targeting of AuNPs is performed by conjugating ligands with high-affinity potential for surface molecules highly expressed on tumor cells or TME. Examples of such ligands are peptides, small molecules, oligosaccharides, and antibodies. These ligands specific to different tumor targets can be functionalized onto AuNPs for improved tumor targeting and cellular localization [65].

Besides acting as nanocarriers, AuNPs can exhibit photothermal effects on irradiation, a two-edged sword strategy to eliminate tumor tissues. In addition, their photon emission potentials and autofluorescence abilities, when irradiated, could be applied as imaging agents and biosensors for cancer detection. AuNPs have been employed in several anticancer strategies such as targeted therapy, gene therapy, diagnostic and imaging, improved radiation therapy, enhanced chemotherapy, immunotherapy, photothermal therapy, and PDT [64,65,66,67]. The subsequent sections will emphasize on AuNP-mediated PDT in cancer and CSCs treatments.

### 5.1. Gold Nanoparticle and Photodynamic Therapy

The efficiency of PDT is primarily regulated by the light absorption/penetration of PS and the generation of ROS. Howbeit, classical PSs are hydrophobic, lack specificity, and have low absorption potential, thereby limiting the clinical use of PDT for deep-seated tumors. These setbacks can be overcome through a nanocarrier. Studies have demonstrated that AuNPs can improve the photodynamic therapy effectiveness of various PSs. The conjugation of PSs to AuNPs has numerous benefits, such as promoting the internalization of PS in the cancer cells increasing the quantum yield of PS through its surface plasmonic resonance activity following light activation, increasing the solubility of the PS, and the possibility of attaching biomolecules to the PS-AuNP for direct PDT targeting. AuNPs can also convert light to heat, promoting the thermal destruction of targeted tumor cells [64,68,69]. In a recent study, a PS AlPcS_4_Cl was conjugated onto an AuNP to evaluate its cytotoxic impacts on lung cancer metastasis. The study showed that PS-AuNP-mediated PDT limits lung cancer’s migratory and invasive potential, inhibits the cell cycle, and decreases lung cancer proliferation [70]. In addition, a study by Chi and colleagues (2020) evaluated the PDT effects of 5-aminolevulinic acid (5-ALA)-AuNPs conjugates on epidermoid carcinoma cells. Their reports showed significant singlet oxygen production and increased cytotoxic cell death with the nanoconjugate compared to the free PS [68].

### 5.2. Gold Nanoparticles Bioconjugation Strategies

Gold NPs unique surface chemistry enables easy bioconjugation with different biomolecules, such as antibodies, aptamer, peptides, polymers, small molecules, and so on, for specific cell targeting, the delivery of anticancer agents to the required sites, and to prolong the drug circulation time at the targeted sites [67]. Surface conjugation is usually employed to enhance the area-to-volume ratio/drug loading capacity, minimize biodegradation, increase solubility, enhance tumor localization, and actively target tumor cells [69,71].

Bioconjugation involves the attachment of biomolecules to NPs through chemical adsorption or physical adsorption to make them stable, functional, and biocompatible. It offers distinct and enhanced features for an effective anticancer drug-delivery system. To enhance the bioconjugates’ stability, increase their circulation time and escape biodegradation, they are often decorated or capped with polymers and capping agents such as polyethylene glycol (PEG), polyvinylpyrrolidone (PVP), polyethyleneimine (PEI), cetyl trimethyl ammonium bromide (CTAB), hyaluronic acid and so on. These decorating agents can be modified with different bifunctional ligands serving as linkers for further chemical interaction of biomolecules [72,73].

Chemical adsorption involves the chemical interactions between the biomolecules and the AuNPs through covalent bonds such as thioester bond, amide bond, hydrazone bond, disulfide bond, bifunctional linkers, adapter molecules (e.g., biotin and streptavidin) and click chemistry (e.g., EDC (N-(3-Dimethylaminopropyl)-N’-ethyl carbodiimide hydrochloride)-NHS (N-hydroxysuccinimide). Chemical adsorption has several advantages, including reducing activity loss, high stability and solubility, prolonged circulation, increased retention in tumor cells, being highly specific and selective for tumor cells, and improved treatment efficiency [69,74,75].

Physical adsorption involves a non-covalent attachment between the biomolecules and AuNPs via electrostatic, hydrophobic, and van der Waals attraction. This type of conjugation is easy, and surface chemistry functionalization is not needed [69,71]. However, physical adsorption methods are associated with limitations such as weak bonding, low stability, biomolecule overloading causing loss of loosely bound antibodies, and the binding is not specific, resulting in random orientation [76].

### 5.3. Impacts of Gold Nanoparticle Physiochemical Properties on PDT

The physicochemical properties of nanoparticles, such as size, shape, surface chemistry and addition of targeting molecules, are extensively explored to enhance the nanoparticle biological interactions such as pharmacokinetics and biodistribution. The physiochemical properties can significantly affect their intracellular action and therapeutic efficiency. Consolidated guidelines have been made for rational nanoparticle design; however, further optimization for each cancer type is required [74,75,77].

Nanoparticle Size: the size of nanoparticles is an essential parameter in examining their intracellular and therapeutic impacts. Usually, nanoparticles of <5 nm are not used for drug-delivery applications due to their renal clearance and rapid circulation time, although ultra-small inorganic nanoparticles are widely used for imaging applications due to their minimal systemic toxicity. Nanoparticles >5 nm to about 100 nm with near to neutral surface charge are considered to be suitable for passive tumor localization as a result of their long-lasting circulation time and ability to escape attack by the immune system [77].

Nanoparticle Shape: Another vital parameter that can influence the efficiency of NPs is the shape. Generally, spherical NPs are considered to demonstrate rapid cellular uptake rate and are suitable as therapeutic nano delivery systems among the various shapes of NPs. Spherical NPs face less membrane hindrance during endocytosis when compared with other shapes [78]. A study conducted by Chithrani and Chan (2007) showed that the size and shape of transferrin-coated AuNPs significantly impact their cellular internalization and exocytosis kinetics. The cellular uptake of the transferrin-coated AuNPs was through receptor-mediated clathrin-dependent endocytosis and was associated with the size and shape of the nanoparticle. In contrast, the cellular efflux of the nanoparticle was basically size-dependent. They also observed that rod-shaped NPs had less cellular uptake and higher exocytosis when compared to spherical-shaped nanoparticles [79]. Gamaleia and colleagues conjugated hematoporphyrin and gold nanoparticles for PDT and examined the effects of different diameters. The study showed that larger NPs allowed for more PS to be delivered to the tumor cells [80]. Furthermore, Khaing Oo and coworkers demonstrated the effects of AuNPs size on PDT. Their findings showed that AuNPs with larger sizes are associated with higher ROS production with PS due to increased dispersion of electromagnetic field surrounding the particles compared to those with smaller sizes [81].

Nanoparticle Surface chemistry: Surface functionalization of AuNPs is considered a suitable strategy for improving biocompatibility and pharmacokinetics for efficient bio-interaction. The surface chemistry regulates the physicochemical features of AuNP surface, which include solubility, electrochemical strength, and homogeneity in solution [82]. Surface characteristics govern the intracellular activities of the AuNPs, cell membrane adsorption strength and uptake, immune response, and in vivo internalization [83]. Suitable surface functionalization/modification of AuNPs can significantly enhance their pharmacokinetics and intracellular interactions. The surface chemistry of AuNPs can be achieved by modifying its surface with coating agents such as protein, peptide, polymers, surfactants, or other agents [84].

Among the several coating agents, polyethyleneglycol (PEG) coating (PEGylation) is the most common NP surface functionalization. The PEG molecules form a solubility sheath on the surface of the NP, which enhances the stability and pharmacokinetics of the NPs as well as prevents non-specific protein attachment and immunogenic attack. In addition, PEGylation facilitates cell internalization more than the non-PEGylated AuNPs [72,85]. Gold NPs functionalization with PEG has been applied in PDT. Surface functionalized AuNRs with mPEG-SH were conjugated with PS AlPcS4. The findings showed that upon low/high PDT application, the free PS was not toxic to non-malignant cells (Shi et al., 2014).

## 6. Gold Nanoparticle Bioconjugates for Active Targeted Photodynamic Therapy for Cancer and Cancer Stem Cells

AuNPs conjugated with PS improve the efficiency of PDT by overcoming the limitation of conventional PDT for the treatment of deep-seated tumors [73]. The application of AuNPs allows surface functionalization with multifunctional ligands to increase the affinity toward cancer cells. The targeting ligands regulate the preferential interaction between the nanoparticles and specific biomolecules highly expressed on the surface of cancer cells/CSCs, sparing the normal cells and consequently increasing cellular uptake [76]. Gold NPs can be modified with biomolecules that have affinities to the various surface and intracellular receptors expressed by cancer cells and CSCs. Nano-cargos employed in drug-delivery systems have the potential to deliver anticancer agents to the CSC niche and improve the treatment effect more than the free anticancer agents. This is achieved by targeting the various biomarkers, signaling pathways, and TME that promote self-renewal, differentiation of CSCs, proliferation, drug resistance [34].

### 6.1. Actively Targeting Biomarkers

Cancer and CSCs can be differentiated from normal cells based on their specific biomarkers. Several biomarkers are associated with cancer/CSCs, as highlighted in Table 2. The strong interaction between the biomarkers and their corresponding ligands promotes improved targeting and increases the anticancer agent’s localization and concentration in cancer/CSCs. Therefore, actively targeting the biomarkers of cancer and CSCs provides an excellent target for potential therapeutic measures for eradicating CSCs [34].

In another study, AuNPs were conjugated with a PS C11Pc and an EGFR peptide AEYLR to target EGFR markers expressed in lung cancer [86]. The study showed improved PDT effects, no dark toxicity, and the efficiency of a peptide as an excellent option for the delivery of AuNP-PS to tumor cells. To improve the target specificity of PDT to cervical cancer cells, Yu and colleagues conjugated transferrin into the AuNP coated with poly (styrene-altmaleic acid) (PSMA), combined with Methylene blue (MB) as a PS via the EDC/NHS reaction. The synthesized nano-bioconjugates (Tf-AuNP-PSMA-MB) showed enhanced affinity and a 2-fold improvement in PDT toward cervical cancer cells compared to normal cells and the free MB [88].

In addition, a study conducted by Stuchinskaya et al. developed a multicomponent anticancer agent consisting of phthalocyanine PS, anti-HER2 antibody, PEG and AuNP for active targeted PDT of breast cancer cells. The anticancer agent showed enhanced PDT through an increase in ROS generation. The findings demonstrated that the conjugation of PSs, AuNPs, and biomolecules as in one entity preferentially targeted HER receptors expressed breast cancer cells [87]. Mangadlao and coworkers conducted another study that employed actively targeting biomarkers. In this study, prostate-specific membrane antigen (PSMA) highly overexpressed in prostate cancer was actively targeted using PSMA peptide combined with AuNP for targeted delivery of silicon phthalocyanine (Pc4) PS. The findings showed increased cellular internalization and significant cytotoxic cell death [89].

Furthermore, neuropilin-1 receptor (NRP-1), a biomarker expressed in glioblastoma cells, was targeted using a nanosystem consisting of a pyropheophorbide-a (Pyro) PS, a peptide (KDKPPR) and a PEGylated Au nanorod (AuNR). The results showed that the nanosystem is an efficient target for drug delivery and enhanced PDT [90]. Liu et al. (2021) synthesized a new PS chlorin e6-C-15-ethyl ester (HB) having the same excitation wavelength as AuNRs for theranostic application. They developed a nanohybrid of AuNRs conjugated with the HB and a cyclic RGD (cRGD) peptide (HB-AuNRs@cRGD) for active targeting esophageal cancer expressing αvβ3 integrin using single activation light wavelength. They reported that the nanohybrid selectively localized in the tumor cells with increased cellular uptake significantly enhanced its anticancer effect through the increased production of ROS and high heat generation [91]. In a recent study, AuNPs were surface functionalized with 6-mercapto-1-hexanol (MH) and conjugated with protoporphyrin IX (PpIX) and folic acid (FA) (PpIX/FA-MH-AuNP) aimed at targeting FA receptors overexpressed on cervical cancer cells. Findings from the study showed that the nano constructs enhanced photochemical localization and cell death in the cancer cells [92].

As reviewed above, several biomarkers of CSCs have been identified and designed for direct targeted delivery of PS to cancer cells for distinct destruction by AuNP-mediated PDT. However, AuNPs bioconjugates-mediated PDT targeting biomarkers of CSCs are few. For example, in a study, AuNPs were conjugated with an anti-CD133 antibody and a PS (aluminum phthalocyanine tetra sulfuric chloride (AlPcS_4_Cl)) to target CD133 biomarker overexpressed in lung cancer stems. The study demonstrated increased cellular toxicity and cell destruction than the PS alone [59]. In another study, aptamers-CSC13, targeted against prostate CSCs, were conjugated onto gold nanorods. The finding demonstrates efficient targeting and cell death of CSCs, confirming the efficiency of AuNPs bioconjugation in eradicating CSCs subpopulation [93].

### 6.2. Actively Targeting Tumor Signaling Pathways

Several studies have evaluated the impact of AuNP bioconjugates on cell signaling pathways in different cancers, as depicted in Table 3. Chi et al. (2020) assessed the effects of 5-aminolaevulinic acid (5-ALA)-AuNPs-mediated PDT on cutaneous squamous cell carcinoma cells and evaluated signaling pathways. The 5-ALA-AuNP conjugates significantly impact the STAT3 signaling pathway as depicted with reduced expression of STAT3 and Bcl-2 and high expression of Bax in the cutaneous squamous cell carcinoma cells the PDT with the free 5-ALA. In addition, the treatment of 5-ALA-AuNPs improved PDT efficiency by reducing cell invasion and migration potential and Wnt/β-catenin signaling effects in the tumor cells [68]. In another report, the phosphatidylinositol 3-kinase (PI3K) inhibitor drug was conjugated with Au nanorod (NR) to investigate its ability to inhibit the PI3K/Akt Pathway and related pathways in breast cancer cell lines. The reports showed the nanoconjugates significantly altered the transcription factors and the protein expression levels of PI3K involved in several pathways associated with cell differentiation and proliferation, cell death signaling, and cell cycle arrest than the single components [94].

In addition, a study conducted by Bhowmik and Gomes (2017) demonstrated that the coupling of AuNP with a cytotoxic protein NKCT1 (AuNP-NKCT1) might promote breast cancer cell death via the estrogen receptor pathway by the deactivation of CDK4 and PI3K/Akt, ERK1/2 and p38, MAPK signaling pathway having significant inhibition on NF-κB, and decreasing the action of MMP9 [95]. Furthermore, Balakrishnan and coworkers coupled quercetin to AuNPs (AuNPs-Qu-5) to evaluate the effects of nano-bioconjugates on apoptosis and its associated signaling pathways on breast cancer cell lines. The study revealed a significant increase in apoptotic bodies with high nuclear condensation, high expression of pro-apoptotic proteins (Bax, Caspase-3), and low expression of the anti-apoptotic protein (Bcl-2) following AuNPs-Qu-5 treatment. In addition, EGFR and its associated signaling components PI3K/Akt/mTOR/GSK-3β were inhibited by the nanoconjugates than the free quercetin [96]. In a recent study, AuNP nanoparticles were conjugated with curcumin and paclitaxel (Au-CP) to examine its anticancer effect on triple-negative breast cancer cell lines and delineate the signaling genes involved. The report showed considerable inhibition of VEGF, CYCLIN-D1, and STAT-3 signaling genes and high expression of the apoptotic Caspase-9 gene [97]. Other anticancer agents for actively targeting CSCs signaling pathways are summarized in Table 4.

### 6.3. Actively Targeting Tumor Microenvironments

The TME consists of cancer cells, the extracellular matrix (ECM), immune and stromal cells (such as tumor-associated macrophages (TAM), cancer-associated fibroblasts (CAF), lymphocytes, myofibroblasts, and endothelial cells), mesenchymal cell populations, exosomes, hypoxia, low pH, matrix metalloproteases (MMP), nutritional deficiencies, and angiogenesis, which mutually work together to promote cancer formation and progression and drug resistance [32]. Targeting the TME or its different elements has become an essential approach to enhancing cancer therapy, especially photodynamic therapy (PDT). AuNPs bioconjugates can serve as a carrier for the delivery of anti-TME agents to the target TME via passive and active targeting by hindering the development of TME and inhibiting the formation and metastasis of cancer/CSCs TME that is formed already [108]. In this section, we will lay more emphasis on AuNP bioconjugates targeting hypoxia, low pH, and MMP, as shown in Table 5.

#### 6.3.1. Targeting Hypoxia

Hypoxia is a general characteristic of the TME that emerges due to increased cancer growth rate surpassing the oxygen supply and reduced blood circulation due to the development of weak vasculature tumors. It has been shown that hypoxic tumor TME significantly promotes cancer formation, proliferation, and invasion [117]. The efficiency of PDT is limited by hypoxia since the conventional PDT is oxygen-dependent to produce lethal ROS. Hence, targeting hypoxia in the TME through oxygen-independent mediated PDT has emerged as an effective tool to improve PDT. A recent review by Lou-Franco and a coworker highlighted that AuNPs possess some enzymatic activities such as reductase, peroxidase, catalase, oxidase, glucose oxidase, and superoxide dismutase, which can be leveraged to alleviate hypoxia [118]. In one study, an oxygen self-generated nanocarrier system consisting of nano metal-organic frameworks (nMOFs), AuNPs (Au@ZIF-8), and Ce6 PS was developed to mitigate the hypoxic TME and enhanced PDT. In this system, the Au@ZIF-8 utilized its catalase potential by catalyzing the high hydrogen peroxide expressed in the TME to produce oxygen to alleviate tumor hypoxia, increasing ROS production with strong cytotoxicity and tumor cell death [109].

Furthermore, to ameliorate tumor hypoxia and enhance PDT, two nanometals made up of Au and rhodium (Rh) (Au@Rh) with catalase activities were developed as nanoenzyme. The Au@Rh is conjugated with ICG and PS decorated with the cancer cell membrane (CM) to form a nanosystem (Au@Rh-ICG-CM). The nanoconjugates catalyzed the hydrogen peroxide in the TME to oxygen through the action of Au@Rh to increase ROS production for enhanced PDT. The decoration with tumor CM allows selective targeting through homology adsorption [110].

In another approach to increase the oxygen level in the hypoxic TME, a nanoplatform made of core-shell gold nanocage and manganese dioxide (AuNC@MnO_2_, AM) was designed for oxygen-boosted immunogenic PDT against metastatic triple-negative breast cancer (mTNBC). The nanoplatform was sensitive to the low pH TME and also served as TME responsive oxygen generator upon light activation to produce ROS. In the presence of a hypoxic TME, the MnO_2_ shell is degraded by the low pH and hydrogen peroxide present in the TME. The degraded MnO_2_ results in the excessive generation of oxygen, thereby enhancing the PDT effect of the nanoplatform following laser irradiation. This approach demonstrated that the nanoplatform alleviates hypoxia at the primary tumor site and induces immunogenic cell death (ICD) with systematic anticancer responses against mTNBC [111].

In a similar strategy, Yin and coworkers constructed a responsive nanoenzyme oxygen generator for enhanced PDT and magnetic resonance (MR) imaging with a smart switch modulator. The nanoenzyme consists of gold nanoclusters (AuNCs) embedded in mesoporous silica and encapsulated with MnO_2_, serving as a switching shield shell (AuNCs@mSiO(2)@MnO_2_). In normal physiological conditions, the nanoenzyme inhibits ROS production by turning off PDT and MR. While in low pH TME, the nanoenzyme reacts with hydrogen peroxide resulting in the degradation of MnO_2_. The MnO_2_ degradation results in massive oxygen production, which in turn improves PDT and MR. The study showed that the nanoenzyme could serve as a potential nanoprobe for efficient theranostic application [112].

#### 6.3.2. Targeting pH in the TME

Low pH is one of the characteristic features of the TME, and it can be employed to activate responsive nanodrugs that release their contents into the tumor site in a pH-dependent manner. For example, citraconic amides can undergo hydrolysis to form primary cationic amines upon a change in pH. Conjugating citraconic amides on NPs can result in the interaction of cationic and anionic nanoparticles leading to nanoparticle aggregation that could be employed for photoacoustic imaging, increasing the signal and preventing efflux [108].

For example, Li and coworkers developed a nanosystem based on the aggregation of AuNPs mediated by citraconic amides as a diagnostic strategy. The nanosystem consists of AuNPs modified with 4-(2-(5-(1,2-dithiolan-3-yl)pentanamido)ethylamino)-2-methyl-4-oxobut-2-enoic acid (LSC) containing a citraconic amide moiety and a cyclic RGDyK peptide coupled with 16-mercaptohexadecanoic acid (c(RGDyK) MHDA) enabling selective and active tumor targeting of αvβ3 integrin antigens overexpressed in the tumor [113]. The results from the study showed optimum and enhanced photoacoustic imaging due to the prolonged circulation time of the nanoconstruct and increased cellular uptake at the tumor site.

Similarly, this approach has been utilized to improve the efficiency of PDT and photothermal therapy (PTT). Liu and colleagues 2018, developed a pH nano-responsive system consisting of mesoporous silica-coated gold nanorods (AuNR@mSiO), loaded with ICG PS and with 2,3-dimethyl maleic anhydride (DMA)-functionalized chitosan oligosaccharide-block -poly (ethylene glycol) polymer (CS(DMA)-PEG) as a pH-responsive polymer. Upon reaching the acidic pH, the amide bonds between CS and DMA are detached, exposing the RLA ([RLARLAR]2) peptide to promote cellular and mitochondrial localization. While the activation of light results in the increased generation of ROS and heat, consequently enhancing the efficacy of PDT and PTT [114].

#### 6.3.3. Targeting Matrix Metalloproteases

Matrix metalloproteases (MMP) belong to the class of zinc-binding metalloproteinases. They are involved in the catalytic destruction of ECM, facilitating rapid tumor cell migration out of the local tumor into the adjacent tissue. In addition, MMPs are overexpressed in the TME and enhance tumor proliferation, invasion, and progression to form metastasis [119]. MMP-2 (Gelatinase A) and MMP-9 (Gelatinase B) are known to significantly contribute to tumor invasion and progression through their proteolytic actions. The substrates of these proteinases can be used to preferentially target nano-bioconjugates to the TME. For instance, Hu and coworkers (2017) developed AuNPs decorated with two peptides that include MMP-2 substrate CPLGVRGDDRGD (peptide 1) and CKKKLVFF (peptide 2). The nanosystem AuNPs@pep1/pep2 includes the substrate peptide PLGVRGDD to identify the presence of MMP-2, an RGD motif for tumor detection, while peptide two functions as an assembly-activated scaffold. High expression of MMP-2 in the TME resulted in the cleavage of peptide one on AuNPs@Pep1/Pep2, consequently the activation of peptide two resulting in the self-aggregation of AuNPs at tumor sites. The aggregated AuNPs increase cellular uptake and improve treatment efficiency, displaying a high thermal imaging signal compared to the control groups with non-cleavable peptides by MMP-2 and without peptide 2 [115].

In another report targeting MMP in the TME, a nanotheranostic system consisting of Au nanostar (AuNS) surface functionalized with bovine serum albumin (BSA), embedded with R-780 (I) iodides PS, and decorated with MMP2 polypeptides (Ac-GPLGIAGQ) (AuNS@BSA/I-MMP2) was designed for concurrent diagnosis and treatment of lung cancer cells. Findings from the study showed that the nanotheranostic system selectively targeted the tumor tissue, suppressed tumor proliferation, and significantly reduced the tumor size by 93% more than the control group without the nanoconstruct [116].

In summary, Figure 2 highlights the potential therapeutic applications of AuNP bioconjugates for active targeted PDT, directed at cancer and cancer stem cell signaling pathways, TME, and biomarkers for efficient tumor elimination.

## 7. Current Limitations and Future Perspective of Gold Nanoparticles Application

There are no doubts that the application of AuNPs has demonstrated excellent features for the management of cancers; however, issues regarding safety are still a call for concern. The inadequacy of proper safety guidelines for the application of NPs in cancer treatment poses a major limitation. Immunomodulators and improved surface functionalization measures could eliminate the potential toxicity/risk impacts [120,121]. Nanoparticles are rapidly taken up by cells due to their minute size. This could result in a high concentration, prolonged localization, cellular impairment, and, consequently, increased toxicity in the body. The combinations of several active bioconjugates in nanoparticles and their likely physiological effects make it very challenging during standard drug evaluation, limiting its progress in medical applications [122]. Nanoparticles are commonly applied as delivery cargos, and not much is known about the impacts of NPs post-delivery. Any promising treatment modality using NPs as a delivery system should stipulate their possible clearance/elimination from the body. The lack of such stipulation makes the application of NPs uncertain in cancer management, particularly in clinical trials and other applications [121,122,123].

Nanoparticles of less than 6 nm are readily excreted through the kidney, while bigger sizes are picked up by macrophages and Kuffer cells for elimination by the liver and spleen. Studies have observed prolonged retention time with nanomaterials in the body before being removed by exocytosis. However, the interval before removal from the body poses a serious threat with regard to possible long-term toxicity implications. Hence, evaluating the implication of poor elimination time would provide clarity on the impact of this technology in the management of cancer [121]. In addition, it is crucial to study the dosimetry of PDT [124] and also to take into consideration that AuNPs are not biodegradable, as this could consequently affect the pharmacokinetics [64,123].

## 8. Conclusions

Past years have witnessed significant advancement and achievement in cancer treatments due to early screening, diagnosis, and the emergence of new therapies. Consequently, cancer patients have also experienced better and improved treatment outcomes. Despite this advancement, the mortality rates each year is still a concern due to the development of drug resistance and cancer relapse. A critical factor in cancer relapse is the presence of CSCs, a small population of tumor cells with self-renewal and proliferation potentials. Though several studies and anticancer agents still focus on reducing tumor size by targeting cancer cells, recent studies are now focusing on attacking CSCs. For efficient eradication of CSCs, the identification and characterization of CSCs markers of various tumor tissues are crucial. These markers can be tailored to therapeutic targets. To date, different markers such as CD44, CD90, CD133, CD271 ABCB5, and ALDH1 have been detected and characterized on several tumors. The quest for alternative therapeutic modalities aimed at targeting cancer cells and CSCs directly with increased efficiency is currently ongoing.

PDT provides a non-invasive and direct treatment approach against cancer. However, PDT is ineffective for the eradication of CSCs due to the PS’s non-specificity and off-target toward CSCs. In recent times the application of nanotechnology has been employed to overcome these limitations. Nanotechnology has also been utilized to identify, characterize, and eradicate CSCs, and has shown optimum efficiency compared to present techniques. Functionalization of AuNPs with CSCs biomarkers, inhibitors of signaling pathways, and components of the TME offers a more target-specific strategy for eliminating CSCs and, consequently, tumor cells.

Most importantly, caution should be taken, as excessive functionalization could also result in undesirable adverse effects. For effective application of AuNP bioconjugates active targeted PDT to take full translation into clinical practice, several challenges need to be tackled, such as long-term effects in the body, biocompatibility and toxicological reaction, stability, etc., reproducibility and reliability of production techniques, bulk manufacturing, and regulatory issues. Therefore, the need to develop a holistic strategy to address the above limitations.

## Figures and Tables

**Figure 1 cancers-14-04558-f001:**
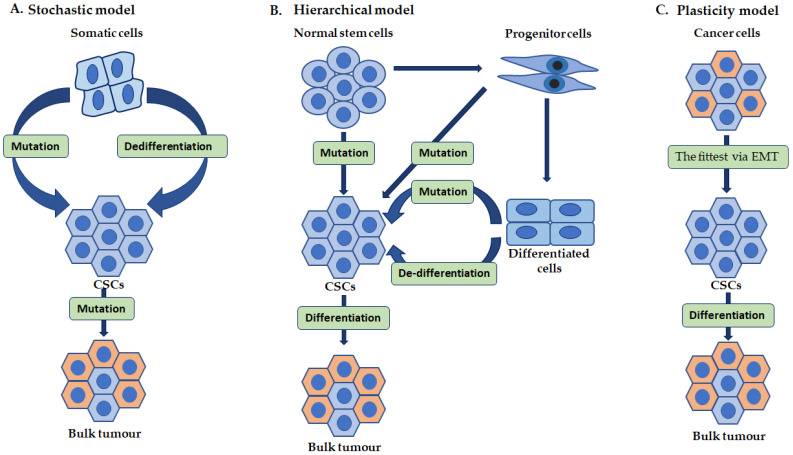
Proposed models of cancer stem cells (CSCs) origin in the emergence of cancer. The development of cancer may originate from somatic cell de-differentiation/mutation, which can self-renew (**A**). CSC may arise from a normal stem cell, a normal progenitor cell, or a normal differentiated cell by mutation/de-differentiation, which turn-on the self-renewal ability (**B**). In addition, tumor cells can develop into CSCs through the EMT (**C**).

**Figure 2 cancers-14-04558-f002:**
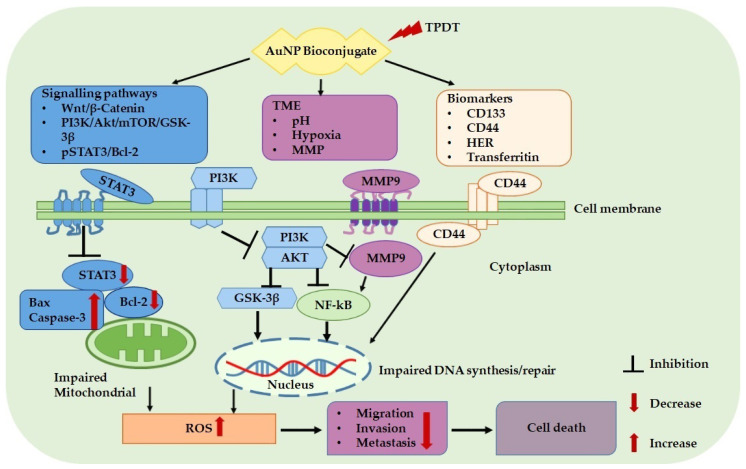
Gold NPs modified with biomolecules that have affinities for cancer/CSCs biomarkers, signaling pathways, and tumor microenvironments with the potential to inhibit invasion, migration, metastasis, and ultimately cell death.

**Table 1 cancers-14-04558-t001:** Highlights of CSC biomarkers and their implications in diagnosis and prognosis of cancer.

**Cancer**	**Marker**	**Diagnostic/Prognostic** **Marker**	**Reference**
Esophagus	CD271	EpCAM+ p75NTR (CD271) is associated with metastasis and vascular invasion. CD90 overexpression is indicative of regional invasion and distant metastasis, and poor prognosis	[13,21]
	ABCG2		
	CD44		
	CD90		
	EpCAM		
Ovarian	CD133	CD24 and ABCG2 are associated with cancer development, metastasis, poor survival, treatment resistance, and relapse	[11,35]
	CD44		
	CD117		
	CD24		
	ABCG2		
Head and neck	CD44	Overexpression of CD44 is associated with a late staged tumor. CD24 is indicative of high proliferation, invasion and drug resistance. CD10 is associated with treatment resistance and relapse.	[15,36]
	CD271		
	CD10		
	CD24		
Lung	CD44	Overexpression of CD44 is associated with advanced-stage cancer. CD166 is usually found in non-small lung cell tumor without nodular metastasis.	[12]
	CD166		
	CD133		
	ALDH		
Melanoma	CD271	CD271 is associated with metastasis and sustains proliferation. ABCB5 induces drug resistance. ALDH1 facilitates cell proliferation, poor prognosis and treatment resistance	[37]
	CD20		
	ABCB5		
	ALDH1		
Breast	ALDH1	ALDH1+CD44+/CD24−/low cells exhibits strong stemness features. ALDH1 is a strong identification marker for breast cancer. Low expression of CD133 is associated with big tumor mass, advanced stage, and vascular invasion. HER2-positive breast cancer expresses a high level of CD133 marker.	[16,38]
	CD44		
	CD133		
	CD24		
Liver	EpCAM	EpCAM, CD133, and CD90 are correlated with tumor invasion, migration, and metastasis and poor prognosis.	[39,40]
	CD133		
	CD44		
	CD90		
Glioma Stem Cells	CD133	Integrin-α6, A2B5, and CD133 are associated with cell proliferation, tumor initiation and drug resistance.	[41]
	CD15		
	Integrin-α6		
	A2B5		

**Table 2 cancers-14-04558-t002:** Gold Nanoparticle Bioconjugates Actively Targeting biomarkers of Cancer and Cancer Stem Cells.

AuNPs-PS Bioconjugates	PS	Biomolecules	Targets	Cancer/CSCs	Reference
EGF Peptide–C11Pc–PEG–AuNPs	C11Pc	EGF peptide (AEYLRK)	EGFR	Lung cancer	[86]
Anti-HER-C11Pc-PEG-AuNPs	C11Pc	Anti-HER antibody	HER	Breast cancer	[87]
Tf-AuNP-PSMA-MB	MB	Transferrin peptide	Transferrin receptor	Cervical cancer cells	[88]
AuNP-PEG-PSMA-1-Pc4	Pc4 Silicon phthalocyanine	PSMA-1 peptide	PSMA receptor	Prostate cancer	[89]
AuNRs@PEG-MI-K(Pyro)DKPPR-OH	Pyro	DKPPR	Neuropilin-1 receptor (NRP-1).	Glioblastoma	[90]
HB-AuNRs@cRGD	HB	Cyclic RGD peptide	αvβ3 integrin	Esophageal cancer	[91]
AuNP-PEG-AlPcS_4_Cl-Anti-CD133	AlPcS_4_Cl	Anti-CD133 antibody	CD133	Lung CSCs	[59]
PpIX/FA-MH-AuNP	PpIX	Folic acid (FA)	FA receptor	Cervical cancer	[92]

**Table 3 cancers-14-04558-t003:** Gold nanoparticle bioconjugates actively targeting tumor signaling pathways of cancer.

AuNPs-PS Bioconjugates	PS/Drug/Molecules	Effects on Signaling Pathways	Cancer Type	Reference
5-ALA-AuNPs	5-ALA	Inhibit STAT3/Bcl-2 and Wnt/β-catenin signaling pathways	Cutaneous squamous cell carcinoma	[68]
PI3K-AuNR	PI3K inhibitor	Inhibit the PI3K/Akt pathway	Breast cancer	[94]
AuNP-NKCT1	Cytotoxic protein NKCT1	Deactivation of CDK4 and PI3K/Akt, ERK1/2 and p38, MAPK signaling pathways	Breast cancer	[95]
AuNPs-Qu-5	Quercetin	Inhibit PI3K/Akt/mTOR/GSK-3β pathways	Breast cancer	[96]
Au-CP	Curcumin and paclitaxel	VEGF, CYCLIN-D1, and STAT-3 signaling genes	Triple-negative breast cancer	[97]

**Table 4 cancers-14-04558-t004:** Other anticancer agents targeting CSCs signaling pathways.

Anticancer Agents	Targeted Signaling Pathways	Cancer Type	Reference
Vismodegib (GDC-0449)	Hedgehog pathway	Multiple basal-cell carcinomas, gastresophageal junction cancer	[98]
Sonidegib	Hedgehog pathway	Triple-negative breast cancer (TNBC)	[99]
Saridegib (IPI-926)	Sonic Hedgehog pathway	Advanced pancreatic adenocarcinoma	[100]
Niclosamide	Wnt/β-catenin	Ovarian cancer, osteosarcoma	[101,102]
Ipafricept (OMP-54F28)	Wnt/β-catenin	Recurrent platinum-sensitive ovarian cancer, advanced solid tumor	[103,104]
Demcizumab (OMP-21M18)	Notch pathway	Metastatic non-squamous non-small cell lung cancer NSCLC	[105]
MK-0752	Notch pathway	Pancreatic cancer	[106]
Alvocidib	STAT3 pathway	Lung cancer colorectal cancer	[107]
Napabucasin	STAT3 pathway	Liver cancer pancreatic cancer	[107]

**Table 5 cancers-14-04558-t005:** Gold nanoparticle bioconjugates targeting the tumor microenvironment.

AuNPs Bioconjugates	PS/Drug	Tumor Microenvironment	Reference
nMOFs-Au@ZIF-8-Ce6 PS	Ce6	Hypoxia	[109]
Au@Rh-ICG-CM	ICG	Hypoxia	[110]
AuNC@MnO_2_	-	Hypoxia	[111]
AuNCs@mSiO(2)@MnO_2_	-	Hypoxia/Low pH	[112]
AuNP-LSC-c(RGDyK) MHDA	-	pH	[113]
AuNR@mSiO-ICG-CS(DMA)-PEG	ICG	pH	[114]
AuNPs@pep1/pep2	-	MMP	[115]
AuNS@BSA/I-MMP2	R-780 (I) iodides	MMP	[116]

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
