# Peer review of "A Gold Nanoparticle Bioconjugate Delivery System for Active Targeted Photodynamic Therapy of Cancer and Cancer Stem Cells"

_cancers, 2022, doi:10.3390/cancers14194558_

Round 1

Reviewer 1 Report

A Gold Nanoparticle Bioconjugate Delivery System for Active Targeted Photodynamic Therapy of Cancer, And Cancer Stem Cells

This paper report that Gold nanoparticle bioconjugates with photodynamic therapy (PDT) could be a potential approach for targeted photodynamic therapy of cancer and CSCs. This approach has the potential for enhanced drug delivery, selective and specific attachment to target tumour cells/CSCs as well as the ability to efficiently generate ROS.

The manuscript is good which will be good addition in the literature however, there are few queries which need to be addressed as;

1.      Merits of the present study should be compared with the literature.

2.      Language of the manuscripr must be improved.

3.      Sentence/paragraph should not be started directly with abbreviation.

4.      In addition to the above mention references, few more recent/relevant references may be cited as;

https://www.ncbi.nlm.nih.gov/pmc/articles/PMC7311980/

Author Response

  1. Merits of the present study should be compared with the literature.

Response: The authors have added new sections to the manuscript

  1. Language of the manuscript must be improved.

Response: We have checked the manuscript thoroughly and made necessary corrections as recommended.

  1. Sentence/paragraph should not be started directly with abbreviation.

Response: We have checked and made necessary corrections as recommended.

  1. In addition to the above-mentioned references, few more recent/relevant references may be cited as;

https://www.ncbi.nlm.nih.gov/pmc/articles/PMC7311980/

https://core.ac.uk/download/pdf/230212371.pdf.          

Response: We have added references as recommended.

Reviewer 2 Report

A Gold Nanoparticle Bioconjugate Delivery System for Active 2 Targeted Photodynamic Therapy of Cancer, And Cancer Stem 3 Cells is  an interesting study, here I attached the following comments,

1-      Include a little bit more about how nanoparticle size and surface chemistry are impacting photodynamic therapy for cancer and cancer stem cells.

2-      Does the route of administration have any impact on photodynamic therapy?

3-      Add a few advanced techniques that are used for the evaluation of photodynamic therapy after the administration of gold nanoparticles.

4-      What about injected doses for optimum therapy?

Author Response

1. Include a little bit more about how nanoparticle size and surface chemistry are impacting photodynamic therapy for cancer and cancer stem cells.

Response: We have included the impacts of nanoparticle size and surface chemistry on photodynamic therapy, as suggested. See page 10, paragraph 4-7; page 11 and paragraph 1.

2. Does the route of administration have any impact on photodynamic therapy?

Response: we have added this information. See page 6, paragraph 5.

3. Add a few advanced techniques that are used for the evaluation of photodynamic therapy after the administration of gold nanoparticles.

Response: We have appropriately addressed it as recommended on page 7, paragraphs 3-5.

4. What about injected doses for optimum therapy?

Response: We have added a more in-depth summary of injected doses for optimum photodynamic therapy. See page 7, paragraph 1 and 2.

Round 2

Reviewer 1 Report

Accepted in present form